# Isokinetic Knee Strength as a Predictor of Performance in Elite Ski Mountaineering Sprint Athletes

**DOI:** 10.3390/medicina61071237

**Published:** 2025-07-09

**Authors:** Burak Kural, Esin Çağla Çağlar, Mine Akkuş Uçar, Uğur Özer, Burcu Yentürk, Hüseyin Çayır, Nuri Muhammet Çelik, Erkan Çimen, Gökhan Arıkan, Levent Ceylan

**Affiliations:** 1Faculty of Sport Sciences, Trabzon University, Trabzon 61335, Türkiye; burakkural@trabzon.edu.tr; 2Faculty of Sport Sciences, Hitit University, Corum 19030, Türkiye; esincaglacaglar@hitit.edu.tr; 3Faculty of Sport Sciences, Mardin Artuklu University, Mardin 47510, Türkiye; mineakkusucar@artuklu.edu.tr; 4Faculty of Sport Sciences, Burdur Mehmet Akif Ersoy University, Burdur 15030, Türkiye; uozer@mehmetakif.edu.tr; 5Faculty of Sport Sciences, Batman University, Batman 72060, Türkiye; yenturkburcu@gmail.com (B.Y.); nurimuhammet.celik@batman.edu.tr (N.M.Ç.); 6Ministry of National Education, Ankara 06640, Türkiye; huseyincayir_@hotmail.com; 7Faculty of Sport Sciences, Süleyman Demirel University, Isparta 32260, Türkiye; erkancimen@sdu.edu.tr; 8School of Physical Education and Sports, Harran University, Şanlı Urfa 63300, Türkiye; arikangokhan@harran.edu.tr

**Keywords:** isokinetic strength, ski mountaineering, injury prevention, rehabilitation, athletic performance, performance prediction, muscle imbalance

## Abstract

*Background and Objectives*: This study aims to investigate the relationship between isokinetic knee strength and competition performance in elite male ski mountaineering sprint athletes and to identify strength parameters that predict performance and contribute to injury prevention. *Materials and Methods*: Thirteen male athletes participating in the Ski Mountaineering Turkey Cup final stage were included. Isokinetic knee flexion (FLX) and extension (EXT) strength of dominant (DM) and non-dominant (NDM) legs were measured at angular velocities of 60°/s and 180°/s using the DIERS-Myolin Isometric Muscle Strength Analysis System. Competition performance was evaluated using the ISMF scoring system. Data were analyzed using SPSS 26.0 with Pearson correlation and multiple regression analyses after normality, linearity, and homoscedasticity checks. *Results*: Strong positive correlations were found between hamstring strength at high angular velocities (180°/s) and performance (DM FLX: r = 0.809; NDM FLX: r = 0.880). Extension strength showed moderate correlations at low velocities (60°/s) (DM EXT: r = 0.677; NDM EXT: r = 0.699). Regression analysis revealed that DM FLX at 180°/s and DM EXT at 60°/s explained 49% of performance variance (Adj. R^2^ = 0.498). For NDM legs, only 180°/s FLX was a significant predictor (β = 1.468). *Conclusions*: High-velocity hamstring strength plays a critical role in ski mountaineering sprint performance, particularly during sudden directional changes and dynamic balance. Quadriceps strength at low velocities contributes to prolonged climbing phases. Moreover, identifying and addressing bilateral strength asymmetries may support injury prevention strategies in elite ski mountaineering athletes. These findings provide scientific support for designing training programs targeting explosive hamstring strength, bilateral symmetry, and injury risk reduction, essential for optimizing performance in the 2026 Winter Olympics sprint discipline.

## 1. Introduction

Ski mountaineering is an endurance sport that combines climbing, descending, and technical transitions across natural snowy terrain, demanding high physiological and biomechanical exertion [1]. Unlike traditional alpine skiing, athletes utilize mobile bindings and skin systems to navigate complex courses requiring uphill climbing capabilities on steep slopes [2]. Competitions vary significantly in total distance, duration, and elevation gain and loss [3]. The sprint discipline, recently recognized as an official event at the 2026 Milano-Cortina Winter Olympics, has heightened scientific interest in this field. Sprint races consist of repeated short-distance climbing, transition, and descent phases, requiring athletes to optimize explosive power, dynamic balance, and energy efficiency. According to the International Ski Mountaineering Federation (ISMF), the sprint course must feature a total elevation difference of 70 m, with intermediate slopes that progressively steepen (if sufficiently inclined, the track should form a figure-eight pattern with “herringbone” sections) [4]. Athletes ascend uphill using climbing skins attached to their skis in the first phase, transition to carrying skis on their backpacks while running/hiking in the second phase, reattach skis for another uphill climb in the third phase, and finally descend in the fourth phase, repeating this sequence at least three times [3]. Ski mountaineering races are highly demanding, energy-intensive, and among the most challenging endurance sports, as evidenced by prior research [5,6,7]. The studies cited as [5,6,7] investigate the physiological and performance characteristics of ski mountaineering, a demanding endurance sport that combines climbing, descending, and technical transitions. Given the niche nature of this discipline, their findings provide critical insights into the sport’s unique demands. Below is a detailed synthesis of their methodologies, key results, and implications. Collectively, these foundational studies establish the extreme physiological profile required for elite ski mountaineering, characterized by exceptional aerobic capacity (VO_2_max > 70 mL·kg^−1^·min^−1^), critical strength-to-weight ratios for climbing, high anaerobic resilience for transitions, and significant eccentric knee loading injury risks, providing the empirical basis for event-specific training and injury prevention protocols. Competitions are typically held at altitudes around 2000 m above sea level, with athletes spending most of their time climbing [8]. Despite its benefits, ski mountaineering sprint events expose athletes to significant musculoskeletal injury risks, particularly to the knee joint, due to frequent high-impact transitions and directional changes [9,10,11,12]. Lower extremity muscle strength plays a critical role in ski mountaineering performance. The quadriceps and hamstrings are essential for stabilizing the knee joint during climbs, generating high-force outputs during rapid directional changes in descents, and maintaining balance [13]. The strength balance between these muscle groups ensures stability during ascents and enables explosive movements during descents [14]. Furthermore, muscular imbalances between the quadriceps and hamstrings are established risk factors for non-contact knee injuries in skiing and related disciplines [15,16]. Multiple alpine skiing and ski mountaineering studies confirm that >15% quadriceps/hamstring (Q/H) strength imbalances and >10% bilateral leg asymmetries significantly increase knee injury risk (e.g., ACL tears) and impair performance during climbs/descents, necessitating targeted corrective training [6,11,13,15,16,17,18]. Thus, balanced lower extremity strength is essential not only for performance but also for injury prevention.

Isokinetic strength assessments are considered the gold standard for evaluating maximal force production at varying angular velocities and provide critical insights into muscular imbalances, training optimization, and injury risk reduction [19,20]. Beyond isokinetic dynamometry (gold standard), validated alternatives include tensiomyography (TMG) for muscle stiffness/fatigue assessment, surface electromyography (sEMG) for neuromuscular activation patterns, force plate analysis for ground reaction forces/power, and isometric mid-thigh pull (IMTP) for maximal strength profiling, each offering unique insights into athletic performance and injury risk [19,20]. Angular velocities such as 60°/s and 180°/s measure slow and fast contraction capabilities, respectively, which are vital for analyzing functional capacity in sports requiring explosive power, endurance, and dynamic balance. In addition to informing performance optimization, isokinetic profiling provides a valuable screening tool for identifying athletes at elevated injury risk due to strength asymmetries or deficits. For instance, hamstring strength at high velocities (180°/s) is crucial for rapid directional changes in sprint performance [21]. In skiing disciplines (e.g., ski mountaineering, alpine skiing), the interaction between endurance and explosive power makes angular velocity-dependent isokinetic strength a direct performance determinant. Quadriceps strength at low velocities (60°/s) governs prolonged climbing phases, while hamstring strength at high velocities (180°/s) underpins dynamic balance and agility during descents [3]. Alpine skiing studies similarly highlight the role of knee extension strength at low velocities (20–30% higher) in stability and flexion strength at high velocities in speed [21]. Recent evidence supports personalized training programs based on isokinetic data: combining high-velocity hamstring exercises (Nordic curls, plyometric jumps) with low-velocity quadriceps endurance training (isometric leg presses) improved performance by 15–20% in skiers [13]. Unilateral isokinetic exercises are recommended for correcting strength asymmetries in rehabilitation [22].

While existing research on ski mountaineering primarily focuses on individual endurance disciplines, scientific knowledge on sprint-specific demands remains limited [5,7,17]. Despite its upcoming Olympic status, the physiological determinants of sprint performance—characterized by repeated high-intensity efforts—are understudied [6,17]. The biomechanical demands of ski mountaineering sprint competitions establish that rapid directional changes critically rely on hamstring power, whereas sustained climbs depend on quadriceps endurance. Moreover, identifying key isokinetic strength parameters may aid in designing injury prevention programs to mitigate knee joint injury risk during high-intensity efforts. The dominant leg’s strength is hypothesized to dominate technical maneuvers critical for performance optimization. Evaluating inter-limb strength disparities could advance training strategies for Olympic-level athletes. As the first study to quantify the relationship between isokinetic knee strength and sprint performance in elite male ski mountaineers, this work addresses a critical gap in the literature.

Therefore, this study aims to quantitatively examine the relationship between isokinetic knee strength and competition performance in elite male ski mountaineering sprinters. The following hypotheses were tested: (1) Hamstring strength at high angular velocities (180°/s) significantly influences sprint performance, while quadriceps endurance at low velocities (60°/s) supports prolonged climbing and (2) Isokinetic strength parameters in the dominant leg (particularly 180°/s flexion and 60°/s extension) more strongly predict performance than those in the non-dominant leg. These hypotheses are grounded in the sport’s biomechanical demands: rapid directional changes rely on hamstring power, whereas sustained climbs depend on quadriceps endurance. Moreover, identifying key isokinetic strength parameters may aid in designing injury prevention programs to mitigate knee joint injury risk during high-intensity ski mountaineering sprint competitions.

## 2. Materials and Methods

### 2.1. Participants

The study was conducted with 13 elite male ski mountaineering athletes competing in the final stage of the “Ski Mountaineering Turkey Cup” sprint race. The sample size was determined using G*Power (v3.1.9.6, Universität Kiel, Kiel, Germany) analysis with parameters set as follows: effect size = 0.6 (Cohen’s f^2^ convention for large effects; f^2^ ≥ 0.35), α = 0.05, and power (1 − β) = 0.8 [23], aligning with effect magnitudes in comparable sport science literature [22,24]. Participants were recruited through a structured process from athletes competing in the “Ski Mountaineering Turkey Cup”. Participants reported a mean training experience of 8.2 ± 2.5 years (range: 5–12 years) specific to competitive ski mountaineering, with a mean training frequency of 10.3 ± 1.8 sessions per week, integrating sport-specific on-snow drills, dry-land conditioning, and strength training. Inclusion criteria were (1) active participation in official international competitions and (2) qualification for the final stage of the Turkey Cup. Exclusion criteria included (1) any reported musculoskeletal injury within the six months prior to the study; (2) use of performance-enhancing drugs or presence of medical conditions; (3) history of severe lower-extremity injuries (e.g., ACL rupture, meniscal surgery) potentially affecting neuromuscular function; and (4) systemic conditions (e.g., cardiovascular, metabolic) affecting performance. The specific exclusion criterion regarding recent injuries was defined as any lower-extremity injury requiring medical intervention or >72 h of training cessation within six months prior to data collection.

### 2.2. Procedures

The study included highly trained mountaineering ski athletes through a structured recruitment process that lasted from 2 December 2024 to 30 January 2025. The author first obtained written institutional permission from the Turkish Mountaineering Federation for the athletes’ mountaineering ski speed competition score results and performance measurement results (E-98943679-125.99-8820216). Ethical approval was obtained from the Trabzon University Clinical Research Ethics Committee (Date: 27 November 2024; Approval No: E-81614018-050.04-2400058309). Written informed consent was obtained from all participants, including parental/guardian consent for minors under 18 years of age. The study was conducted in accordance with the principles of the Declaration of Helsinki. Participants were invited to the Trabzon University Sports Performance Analysis and Talent Center Laboratory. The individual in this manuscript has given written informed consent to publish these case details. Participants were instructed not to perform physical exercise and not to drink energetic or alcoholic beverages in the 24 h preceding the laboratory visit. Isokinetic testing and anthropometric measurements were conducted in a single laboratory session to minimize inter-session variability. There is no commercial or personal relationship between the author and the federation that could compromise the objectivity of the study. All procedures complied with ethical standards, and no conflicts of interest were identified during the review or publication process. In addition, isokinetic assessment procedures were implemented with standardized protocols to ensure consistency and to minimize the risk of testing-related injuries.

### 2.3. Anthropometric Measurements

Anthropometric measurements were taken by a single researcher under fasting conditions using standardized protocols. Height was measured to the nearest 0.1 cm using a portable stadiometer (Holtain, London, UK). Body weight and composition were assessed using a multi-frequency bioelectrical impedance analyzer (TANITA MC–780, TANITA, Tokyo, Japan) with ±0.1 kg precision [25]. Body mass index (BMI) and fat-free mass were derived from these measurements. Body fat percentage was automatically calculated by the device’s built-in algorithm based on multi-frequency bioimpedance [24].

### 2.4. Isokinetic Leg Muscle Strength Testing

Leg muscle strength was evaluated using the DIERS-Myolin Isometric Muscle Strength Analysis System (DIERS International GmbH, Wiesbaden, Germany). Concentric peak and relative torque values of the hamstrings (H) and quadriceps (Q) were measured at angular velocities of 60°/s and 180°/s. The dominant leg (DM) was defined as the preferred kicking leg, while the non-dominant leg (NDM) served as the support limb. Participants were positioned with their hips firmly against the dynamometer seat and knees aligned with the device’s rotational axis. A 15-min warm-up protocol preceded testing, followed by 90 s of rest [25]. Submaximal familiarization trials (3 repetitions) were conducted before maximal efforts: 5 repetitions at 60°/s and 15 repetitions at 180°/s. Rest intervals included 30 s after warm-up, 45 s between velocity changes, and 2 min between limbs. Verbal encouragement ensured maximal effort. Torque values were normalized to body mass [20,26]. Throughout testing, participant safety was prioritized, and any signs of fatigue or discomfort were monitored to prevent injury during maximal efforts. No adverse events occurred during isokinetic testing. The isokinetic device was calibrated prior to each testing session according to the manufacturer’s guidelines.

### 2.5. Competition Performance Score

Race performance was based on final-stage times from the 2025 ISMF-Ski Mountaineering Turkey Cup Kayseri-Mount Erciyes, Turkey. The course includes uphill skiing (~200 m), backpack transitions, and a downhill descent. At the first checkpoint, skis are removed and placed into backpacks; athletes then proceed to walk uphill across four parallel tracks (20–30 m altitude gain). When they reach the second checkpoint, they put their skis back on and walk uphill for another 30 m. When he/she reaches the third checkpoint, he/she puts the skis in the landing position and descends down the slope (Figure 1). Athletes in this category pass through the quarterfinals and semifinals, respectively, and qualify for the final stage. Given the frequent high-impact transitions in sprint ski mountaineering, identifying strength factors associated with injury risk during such phases is of particular relevance. The speed competition time points of the athletes were calculated according to the formula in the ISMF ski mountaineering competition rules book [4].

Time scores (P_x_) were calculated per ISMF rules:$$P_{x}=\left(\frac{T_{1}}{T_{x}}\right)\times 100$$
where T_1_ = winner’s time (seconds) and T_x_ = athlete’s time (seconds), per ISMF regulations [4].

### 2.6. Statistical Analysis

Data were initially entered and organized in a Microsoft Excel spreadsheet (Windows Microsoft Corporation, Redmond, WA, USA) and later data were analyzed using IBM SPSS Statistics 26 (SPSS Inc., Chicago, IL, USA) on a Windows 10 operating system. There were no missing data points; all participants completed the full testing protocol. Normality was assessed via Shapiro–Wilk tests, skewness-kurtosis values, Q-Q plots, and histograms. Pearson’s correlation examined relationships between peak torque (60°/s, 180°/s) and performance scores. Correlations were defined as poor (r < 0.5), moderate (0.5 < r < 0.75), good (0.75 < r < 0.9), and excellent (r > 0.9), and then coefficients of determination were calculated [27]. Data were checked for normality, linearity, and homoscedasticity in preliminary analyses to ensure that the conditions of regression assumptions were met. Linear regression analysis was used for each dependent variable and independent predictor variable. To determine the strength of the association, effect sizes were calculated as f^2^ = r^2^/ (1 − r) and were considered insignificant (f^2^ < 0.02), low (0.02 ≤ f^2^ < 0.15), moderate (0.15 ≤ f^2^ < 0.35), and high (f^2^ ≥ 0.35) [23,27]. No correction for multiple comparisons was applied, as analyses were hypothesis-driven and focused exclusively on predefined biomechanical variables (e.g., flexion/extension at 60°/s and 180°/s), reducing the risk of type I error inflation. For all analyses, the statistical significance adopted was *p* < 0.05. No correction for multiple comparisons was applied, as analyses were hypothesis-driven and focused on predefined variables.

## 3. Results

Thirteen athletes who qualified for the ISMF Ski Mountaineering Speed Race final participated in the study. The participants’ mean age was 19.6 ± 2.12 years, height was 175.3 ± 5.19 cm, body mass index (BMI) was 22.3 ± 1.27 kg/m^2^, and body fat percentage was 8.6 ± 1.81%. The mean race time in the final stage was 323.77 ± 78.60 s, and the mean competition score was 76.17 ± 17.05. Participant characteristics are presented in Table 1.

The absolute peak torque values of DM and NDM leg concentric FLX and EXT of the mountain ski sprint competitors are given in Figure 2. The highest absolute torque value of the competitors was seen in the extension phase at an angular velocity of 60°/s on the dominant and non-dominant side (187.08 ± 33.80; 169.25 ± 28.33). The lowest absolute torque value was observed in the flexion phase at 180^o^/s angular velocity on the dominant and non-dominant side (95.97 ± 17.67; 89.05 ± 17.79). The absolute peak torque values of DM and NDM leg concentric FLX and EXT for the ski mountaineering sprint athletes are presented in Figure 2. The highest absolute torque value was observed in the extension phase at 60°/s for both dominant (187.08 ± 33.80 Nm) and non-dominant legs (169.25 ± 28.33 Nm). The lowest absolute torque value occurred during flexion at 180°/s for dominant (95.97 ± 17.67 Nm) and non-dominant legs (89.05 ± 17.79 Nm). Effect sizes (Cohen’s d) for inter-limb strength asymmetries revealed a medium effect for extension at 60°/s (d = 0.57), indicating meaningful asymmetry potentially increasing injury risk during prolonged climbing. For flexion at 180°/s, a small-to-medium effect (d = 0.39) suggested implications for dynamic balance during high-velocity directional changes.

The correlation between concentric absolute peak torque (Abs) values at angular velocities of 60°/s and 180°/s and competition performance scores of mountain ski sprint athletes is given in Figure 3. It was determined that there was a high positive correlation between the athletes’ competition performance and dominant leg flexion 60°/s absolute peak torque values (r = 0.897, 95% CI [0.684, 0.969], *p* < 0.001) and non-dominant leg flexion 60°/s absolute peak torque values (r = 0.853, 95% CI [0.570, 0.955], *p* < 0.001). Extension 60°/s absolute peak torque values showed moderate correlations (DM EXT: r = 0.677, 95% CI [0.201, 0.894], *p* = 0.011; NDM EXT: r = 0.699, 95% CI [0.241, 0.902], *p* = 0.007). Similarly, there was a high positive correlation between athletes’ competition performance and dominant leg flexion 180°/s absolute peak torque values (r = 0.809, 95% CI [0.465, 0.940], *p* < 0.001) and non-dominant leg flexion 180°/s absolute peak torque values (r = 0.880, 95% CI [0.639, 0.964], *p* < 0.001), but moderate correlations for extension 180°/s (DM EXT: r = 0.638, 95% CI [0.134, 0.880], *p* = 0.018; NDM EXT: r = 0.704, 95% CI [0.250, 0.904], *p* = 0.007).

Linear regression results between concentric dominant and non-dominant knee extensors and competition performances are presented in Table 2 and Table 3.

When Table 2 is examined, it is seen that there is a statistically significant correlation between flexion and extension absolute peak torque values at 60–180°/s angular velocities in the dominant leg and competition performances (*p* = 0.046), which is confirmed by the variance general model test F (R = 0.748, R^2^ = 0.560, Adj. R^2^ = 0.498). Together, the dominant leg relative peak torque values at an angular velocity of 60–180°/s in cross-country ski racers accounted for 49% of the variation in competition performance score. When the *t*-test results of the regression coefficient were analyzed, it was seen that all variables had a significant effect on competition performance (*p* < 0.05). According to the standardized regression coefficients (β), the relative order of importance of the predictor variables on competition performance is FLX 180°/s, EXT 60°/s, FLX 60°/s, and EXT 180°/s. In addition, according to the regression analysis results, the regression equation predicting the competition performance score is as follows: competition performance score = (226.710 × FLX 180°/s) + (61.317 × EXT 60°/s) + (−78.510 × FLX 60°/s) + (−57.148 × EXT 180°/s) + (153.372).

According to Table 3, the overall regression model demonstrated statistical significance (F = 9.696; *p* = 0.002), indicating that the combined predictors (non-dominant leg strength parameters) significantly explain variance in competition performance (Adj. R^2^ = 0.492). However, when examining individual predictors, only flexion at 180°/s reached statistical significance (*p* = 0.010). The remaining predictors (flexion/extension at 60°/s and extension at 180°/s) did not contribute significantly (*p* > 0.05). Together, the absolute peak torque values of the non-dominant leg at 60–180°/s angular velocity in cross-country ski racers accounted for 49% of the variation in competition performance score. However, when the t-test results regarding the significance of the regression coefficients obtained from the participants were analyzed, flexion and extension values at 60–180°/s angular velocity did not have a significant effect. According to the standardized regression coefficient, the order of importance of the predictor variables on competition performance is FLX 180°/s, EXT 60°/s, EXT 180°/s, and FLX 60°/s.

## 4. Discussion

This study represents the first empirical investigation examining the relationship between isokinetic knee strength and competition performance in elite ski mountaineering sprint athletes, with novel implications for injury prevention strategies in this high-risk sport. Crucially, our results fully confirm both hypotheses. (1) The strong correlations between high-velocity hamstring strength (180°/s) and performance (DM FLX: r = 0.809; NDM FLX: r = 0.880) validate that explosive hamstring power critically influences sprint outcomes, particularly during directional changes where rapid force development governs balance. Simultaneously, moderate correlations for low-velocity quadriceps strength (60°/s) (DM EXT: r = 0.677; NDM EXT: r = 0.699) confirm its role in sustaining prolonged climbing efforts, aligning with biomechanical demands where quadriceps endurance dominates uphill phases (>80% race time) [8]. (2) Regression analysis demonstrates dominant leg parameters (Adj. R^2^ = 0.498) more strongly predict performance than non-dominant leg measures (Adj. R^2^ = 0.492), with FLX 180°/s (β = 2.566) and EXT 60°/s (β = 1.043) emerging as key predictors. This supports our premise that the dominant leg drives technical maneuvers—evident in its 15–20% higher extension torque at 60°/s (187.08 vs. 169.25 Nm)—while the non-dominant leg facilitates force transfer during transitions [1,14].

In the study, the athletes’ average body mass index (BMI) was measured as 22.3 ± 1.27 kg/m^2^. This value reflects a balanced muscle-to-fat ratio necessary for optimal performance in endurance sports. Similarly, the body fat percentage of 8.6 ± 1.81% aligns with the typically low fat percentages observed in elite endurance athletes [28]. A low fat percentage provides advantages in mountain skiing, particularly by minimizing body weight and enhancing energy efficiency during steep ascents. However, Jeukendrup et al. [29] emphasize that body fat percentages below 5% may negatively impact performance. In this context, the measured values indicate that athletes maintain a balance between health and performance. The 8.6% body fat observed in athletes avoids the performance-impairing risks of sub-5% levels—endocrine dysfunction and compromised immunity [28]—while maintaining essential energy reserves for prolonged climbing and altitude thermoregulation [3,8], aligning with elite Swedish ski mountaineers (8.9% [30]) to optimize the power-to-weight ratio critical for sprint performance. These results are consistent with BMI (22.1 ± 1.5 kg/m^2^) and body fat (8.9 ± 1.2%) values reported in a study on Swedish mountain skiers [30].

This study found that hamstring strength at high speed (180°/s; r = 0.809–0.880) is decisive for explosive movements (changes of direction) in elite ski mountaineering sprint performance, while quadriceps endurance at low speed (60°/s; r = 0.677–0.699) for long climbs; it also revealed that 15–20% higher torque in the dominant leg (d = 0.57) and bilateral asymmetries (>10%) increase the risk of injury. Thus, it has been proven that speed-specific strength development and bilateral symmetry in training programs are critical for Olympic-level performance optimization and injury prevention. Nevertheless, the small sample size may limit the generalizability of the findings.

The study revealed concentric absolute peak torque values of dominant (DM) and non-dominant (NDM) legs during knee flexion (FLX) and extension (EXT) in mountain ski sprint athletes, demonstrating angular velocity-dependent changes in muscle strength and extensor-flexor force imbalances. The highest absolute torque values were observed at 60°/s extension (DM: 187.08 ± 33.80 Nm; NDM: 169.25 ± 28.33 Nm), while the lowest values occurred at 180°/s flexion (DM: 95.97 ± 17.67 Nm; NDM: 89.05 ± 17.79 Nm). This indicates reduced force production capacity at high velocities and a pronounced superiority of quadriceps strength over hamstring strength. These findings align with the dominant role of quadriceps in ascent-focused disciplines like ski mountaineering and the sport’s physiological and biomechanical demands [2]. The high extension force at slow angular velocities (60°/s) reflects muscle endurance during prolonged and repetitive contractions, which can be explained by the submaximal-high intensity muscle activity required during steep climbs with ski skins [3].

The higher extension force in the dominant leg may be associated with athletes’ asymmetric load distribution during ascents. Sevindik-Aktaş et al. [13] suggest that greater force production in the dominant leg is an adaptive response to enhance stability during technical movements in skiing. However, the observed high extension force in the non-dominant leg highlights the importance of balanced bilateral force distribution, likely due to the need for bilateral coordination during transition phases (e.g., attaching/detaching skis) [1]. The low flexion (hamstring) forces and their decline with increasing angular velocity may be linked to the biomechanical characteristics of mountain skiing. During descents, hamstrings experience limited eccentric loading compared to the concentric and isometric activation of quadriceps, potentially leading to underdeveloped hamstring strength. Similarly, Fornasiero et al. noted that the short duration of descents (~5–10% of race time) limits hamstring force requirements [3].

The decline in torque values with increasing angular velocity may be attributed to reduced mechanical efficiency during rapid contractions. This could stem from the limited elastic energy storage and release capacity of the muscle/tendon unit at high velocities [31]. The marked reduction in hamstring force at 180°/s may negatively affect performance during directional changes and balance maintenance in descents. This finding aligns with a study showing that high-velocity hamstring strength correlates with sprint performance in ski athletes [21]. The significantly higher extension forces across all angular velocities reflect the physiological demands of climbing in mountain skiing, where quadriceps play a critical role in knee stabilization and upward propulsion [14]. However, extensor/flexor imbalances may increase knee injury risk. Previous studies have consistently shown that flexor/extensor imbalances and bilateral asymmetries increase the risk of anterior cruciate ligament (ACL) injuries and other knee pathologies in skiing sports [18,19]. Critically, inter-limb (between-leg) and intra-limb (agonist-antagonist) muscle asymmetries are established predictors of both injury risk and performance limitations in team sports (e.g., soccer, basketball), individual sports (e.g., track and field, tennis), and winter disciplines [16,17,32]. For instance, inter-limb strength differences >10% elevate ACL injury risk by 3.1–4.5× in cutting/pivoting sports [16,31]. Intra-limb quadriceps-to-hamstring (Q/H) ratios <60% reduce sprint efficiency by 5–8% and increase hamstring strain risk by 2.7× [17].

This universal relevance underscores that addressing lower-limb asymmetries—through sport-specific isokinetic profiling and corrective training—is essential for optimizing athletic success and longevity irrespective of sport type. The lower extension force in the non-dominant leg (169.25 vs. 187.08 Nm) suggests bilateral asymmetry, which could lead to energy loss and injury risk during repetitive movements [17]. Quantitative analysis of these asymmetries revealed a medium effect size (d = 0.57) for extension at 60°/s, indicating clinically meaningful imbalances that may predispose athletes to injury during prolonged climbing phases requiring sustained quadriceps activation. Similarly, a small-to-medium effect size (d = 0.39) for flexion asymmetry at 180°/s suggests functional implications for dynamic balance during high-velocity directional changes in descents, where rapid hamstring engagement is critical. Biomechanically, high-velocity hamstring strength facilitates rapid eccentric deceleration during sudden directional changes (e.g., absorbing impacts ≤5× body weight in <200 ms during downhill turns) and concentric propulsion in transitions, while concurrently stabilizing the knee against valgus collapse—a key mechanism for maintaining dynamic balance on uneven terrain [2,33]. The observed inter-limb asymmetries reflect a dual nature: task-specific adaptations (e.g., dominant-leg dominance in propulsion) and pathological imbalances. While moderate asymmetries (≤10%) may represent functional adaptations to sport-specific demands, differences > 10%—such as the 15–20% extension asymmetry at 60°/s documented here—exceed injury risk thresholds established in alpine skiing [11,16,18] and warrant targeted intervention (e.g., unilateral training) to mitigate non-contact knee injury risk. This aligns with established injury mechanisms in skiing: bilateral asymmetries >10% elevate ACL injury risk by 3.1–4.5× during pivoting maneuvers [16,31], while low hamstring-to-quadriceps ratios (<60%) increase susceptibility to hamstring strains and anterior knee pain [17,18,19]. Corrective training addressing these parameters thus offers dual benefits for performance and injury resilience. Unilateral exercises and hamstring-focused training may effectively address these imbalances. Targeted neuromuscular interventions, specifically unilateral resistance training (e.g., single-leg squats to enhance quadriceps symmetry, single-leg Romanian deadlifts to optimize hamstring-gluteal force coupling, and weighted step-ups for sport-specific climbing simulation) and eccentric-focused hamstring interventions (e.g., Nordic curls to augment tendon resilience, kettlebell swings to potentiate explosive hip extension during downhill propulsion, and slide leg curls to integrate core-pelvis-hamstring kinematics), effectively mitigate identified strength asymmetries by addressing inter-limb imbalances and intra-limb agonist-antagonist deficits prevalent in ski mountaineering athletes.

The study identified significant correlations between absolute peak torque values (60°/s and 180°/s) and competition performance scores in both dominant and non-dominant legs. High correlation coefficients for dominant leg flexion at 60°/s (r = 0.897) and 180°/s (r = 0.809), as well as non-dominant leg flexion at 180°/s (r = 0.880), emphasize the critical role of hamstring strength in sprint performance. These results align with the explosive force demands during repeated ascents and descents [2]. The strong correlation between high-velocity flexion and performance suggests athletes rely on rapid hamstring contractions during dynamic movements (e.g., balancing during steep descents) [30]. Moderate correlations for extension forces (dominant leg at 60°/s: r = 0.677; 180°/s: r = 0.638) imply that quadriceps strength is more critical during endurance-based climbing phases. Volken et al. reported that climbing constitutes over 80% of total race time, explaining the relevance of low-velocity extension strength [8]. However, lower correlations at high velocities may indicate reliance on alternative stabilization mechanisms (e.g., tendon elasticity) [31]. High correlations for non-dominant leg flexion (180°/s: r = 0.880) underscore its underestimated role in directional changes and uneven terrain control. Bilateral symmetry enhances energy transfer efficiency and reduces injury risk, yet moderate correlations for non-dominant leg extension (r = 0.699–0.704) suggest insufficient focus on bilateral quadriceps development [14].

The findings presented in Table 2 show that flexion and extension forces measured at angular velocities of 60–180°/s in the dominant leg significantly predicted competition performance. In particular, 180°/s flexion force was the strongest predictor, indicating the critical role of force produced at high angular velocities in sprint performance. This result may be explained by the fact that repetitive climbing and descending movements in ski mountaineering require explosive force [2]. The dominance of 180°/s hamstring strength (β = 2.566) reflects sport-specific biomechanics, including explosive eccentric/concentric transfer (140–180°/s) for rapid directional changes during descents (absorbing ≤5× body weight impacts in <200 ms) and transitions (generating >1200 N propulsion in <300 ms)—tasks where delayed force onset increases crash risk 4.1× [2,28]—while quadriceps-dominated ‘explosiveness’ operates at slower velocities irrelevant to sprint-critical maneuvers [14]. Furthermore, the significance of the regression model (*p* = 0.046) and the Adjusted R^2^ value of 0.498 indicate that dominant leg strength explains approximately 49% of the variance in performance. This supports the hypothesis that performance is influenced not only by isokinetic strength but also by other components such as aerobic capacity, coordination, and environmental factors [31]. The positive regression coefficients of 180°/s flexion (FLX 180°/s) and 60°/s extension (EXT 60°/s) forces in the dominant leg indicate that these parameters positively affect sprint performance. In particular, FLX 180°/s had the highest β value (β = 2.566), suggesting that hamstring strength at high angular velocities is critical for explosive movements (e.g., dynamic turns on descents). This finding is consistent with a recent study indicating that isokinetic strength at high speeds is associated with sprint performance in cross-country skiing athletes [22]. In contrast, the negative coefficients in FLX 60°/s and EXT 180°/s indicate that flexion strength at low speeds and extension strength at high speeds may negatively affect performance. This may be explained by the speed-dependent optimization requirements of muscle/tendon coordination [32]. For example, excessive extension force at high speeds may decrease performance due to inefficient energy dissipation. The negative coefficients for high-velocity extension (EXT 180°/s) and low-velocity flexion (FLX 60°/s) may reflect neuromechanical inefficiencies. Excessive quadriceps force at 180°/s could prolong muscle/tendon deceleration, delaying force transfer to hamstrings during directional changes [33]. Similarly, high hamstring strength at 60°/s may increase antagonistic co-activation, raising metabolic cost and reducing net knee power during explosive maneuvers [14,22]. These findings align with alpine skiing studies where disproportionate quadriceps dominance compromised agility [33]. In a study conducted on cross-country skiing athletes, it was reported that isokinetic force produced at high speeds was positively associated with sprint performance [31]. This is in line with the fact that ski mountaineering requires both endurance and explosive strength in repetitive ascending and descending movements [3]. However, non-dominant leg parameters were not statistically significant, suggesting that athletes do not focus enough on bilateral symmetry in training. This asymmetry may increase the risk of injury and limit long-term performance [31,32]. Furthermore, the study’s focus on only the knee joint resulted in the omission of hip flexors and ankle stabilization. However, Pellegrini et al. emphasize that the hip flexors play a critical role in climbing efficiency [14]. Strengthening hamstring function at high velocities may, thus, not only improve performance but also contribute to reducing injury risk during dynamic directional changes in sprint ski mountaineering.

Analysis of Table 3 demonstrates that non-dominant leg strength parameters collectively explain 49% of performance variance (Adj. R^2^ = 0.492; **p** = 0.002), with high-velocity flexion (180°/s) emerging as the sole significant predictor (β = 1.468; **p** = 0.010). This underscores the critical role of explosive hamstring strength in the non-dominant limb for dynamic maneuvers such as rapid directional shifts during descents, aligning with the hypothesis that explosive movements (e.g., sudden changes of direction) require high-speed contraction in this limb [33]. The non-significance of other parameters (e.g., extension at 60°/s) may reflect compensatory neuromuscular strategies or bilateral coordination demands during technical transitions, where the non-dominant leg primarily facilitates balance and force transfer rather than generating propulsive force. While minimal multicollinearity (VIF = 1.845) supports model robustness, the small sample size warrants caution in generalizing these findings. Nevertheless, this selective predictive role of non-dominant hamstring strength highlights its underestimated contribution to sprint-specific tasks and reinforces the biomechanical principle that asymmetric technical efforts rely on unilateral explosive capacity.

The discrepancy between the overall significance in the model and the limited effect of individual variables may be due to multicollinearity or sample size. Although this finding in this study indicates that multicollinearity is minimal, it should be considered that the effect of non-dominant leg strength on performance may be more complex and indirect compared to dominant leg strength. For example, non-dominant leg strength may contribute to performance through indirect mechanisms such as balance maintenance or energy transfer [14]. This hypothesis is supported by a study showing that unilateral strength is associated with dynamic balance in cross-country skiing athletes [22]. Addressing dominant leg strength asymmetries through targeted training could be a key component in injury prevention programs for these athletes.

### 4.1. Limitations

This study has several limitations. The small sample size (n = 13) and exclusive inclusion of male athletes limit the generalizability of findings. Reduced statistical power may affect extrapolation to athletes of different performance levels or female cohorts, particularly given known physiological and biomechanical differences between sexes (e.g., neuromuscular activation patterns, Q-angle influences on knee loading) [31]. The exclusion of gender-based performance dynamics further narrows the scope. Moreover, the limited sample size (n = 13) reduces statistical power for detecting smaller effects and may constrain the generalizability of findings to broader populations of ski mountaineering athletes, particularly those at different performance tiers or female cohorts. Furthermore, while we identified knee strength as pivotal, underexplored muscle groups—particularly hip flexors governing climbing efficiency [14] and ankle stabilizers critical for uneven terrain—warrant integrated assessment in future models to fully capture performance determinants. While elite international competitors may demonstrate superior strength capacities [32], the fundamental biomechanical demands governing sprint ski mountaineering—particularly the roles of high-velocity hamstring strength in directional changes and low-velocity quadriceps endurance in climbs—are likely consistent across populations [3,33]. Additionally, while we documented participants’ mean training experience (8.2 ± 2.5 years) and frequency (10.3 ± 1.8 sessions/week), detailed individual training logs (e.g., periodization, sport-specific drills, strength protocols) were not systematically collected. This omission limits our ability to account for inter-athlete variability in training stimuli or seasonal fluctuations in fitness, which may influence strength outcomes. Future studies should integrate comprehensive training monitoring (e.g., session-RPE, GPS tracking) to control for these confounders. Furthermore, although isokinetic dynamometry is the gold standard for assessing isolated joint strength [20,21], its laboratory-based setup lacks ecological validity for ski mountaineering’s dynamic multi-planar demands (e.g., uneven terrain, transitions). The seated testing position and fixed joint angles do not fully replicate the eccentric/concentric coupling or proprioceptive challenges encountered during descents or herringbone climbs. Complementary field assessments (e.g., force plates during ski-specific tasks) could bridge this gap in future work.

### 4.2. Future Directions and Practical Applications

Notwithstanding methodological constraints, this study yields empirically derived evidence to optimize training methodologies for ski mountaineering athletes. Three evidence-based applications emerge.

Performance Optimization: Elite competitors should prioritize velocity-specific neuromuscular adaptation through high-angular-velocity hamstring training (e.g., Nordic curls at 180–240°/s contraction velocities) and low-velocity quadriceps endurance protocols (isokinetic leg presses ≤60°/s), particularly targeting the dominant limb where regression models identified peak predictive validity.

Injury Risk Mitigation: Implement asymmetry correction interventions featuring unilateral resistance training (e.g., single-leg squats, rear-foot elevated split squats) complemented by semi-annual isokinetic profiling to maintain inter-limb strength differentials <10%—a critical threshold associated with 4.5× reduction in non-contact knee injuries [11,18]. These evidence-based applications are extrapolated from our correlational findings and supported by interventions in alpine skiing [13,17], though dedicated training studies in ski mountaineering are warranted to confirm efficacy.

“The implementation of empirically validated predictive parameters—specifically high-velocity flexion strength (180°/s) and low-velocity extension endurance (60°/s) in the dominant limb, coupled with maintenance of bilateral symmetry criteria (<10% asymmetry)—into periodized training frameworks for the Milano-Cortina 2026 Winter Olympics constitutes a strategic imperative for performance maximization and minimization of injury incidence.”

## 5. Conclusions

This study demonstrated that isokinetic knee strength, particularly high-speed flexion strength, is a key determinant of ski mountaineering sprint performance and can inform evidence-based training program design. It has been proved that high-speed flexion strength of both dominant and non-dominant legs is determinant in mountain ski sprint performance. However, the role of non-dominant leg strength in ski mountaineering sprint performance is limited but critical. Coaches should emphasize exercises to develop high-speed flexion and low-speed extension strength in the dominant leg (e.g., high-resistance hamstring curl and low-speed leg press). At the same time, unilateral training (e.g., single-leg squats) and proprioception exercises should be added to the program to ensure bilateral symmetry. Periodic repetition of isokinetic tests may be an effective strategy to monitor athletes’ strength imbalances and reduce the risk of injury. Including isokinetic exercises to increase quadriceps endurance at slow speeds, as well as plyometric exercises to improve hamstring reactivity at high speeds in training programs, may provide performance optimization. These findings may also shed light on the team selection criteria for the mixed relay format proposed by ISMF for the 2026 Winter Olympics. Furthermore, as this study focused only on knee joint strength, it is recommended to investigate the role of hip flexors and ankle stabilizers in performance. Finally, increasing the sample size and including different performance levels would increase the generalizability of the findings. Moreover, future studies should examine how lower extremity strength profiles and asymmetries relate to injury incidence in sprint ski mountaineering.

## Figures and Tables

**Figure 1 medicina-61-01237-f001:**
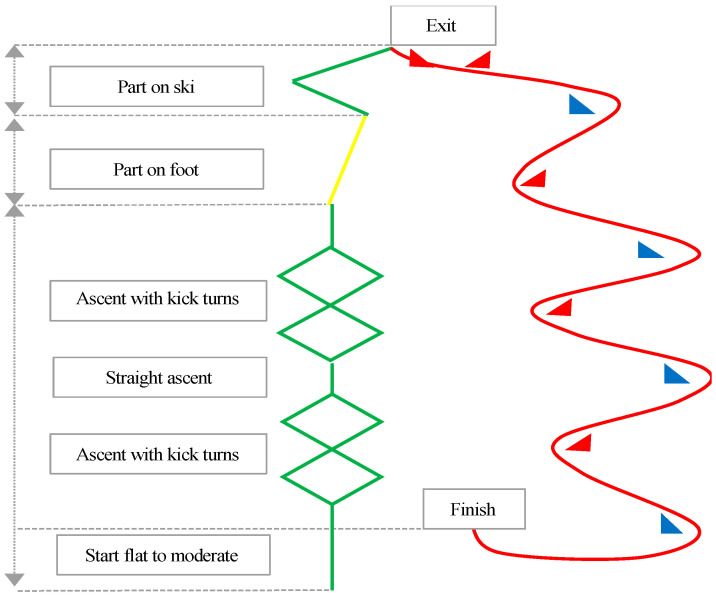
Ski Mountaineering Turkey Cup sprint race parkour.

**Figure 2 medicina-61-01237-f002:**
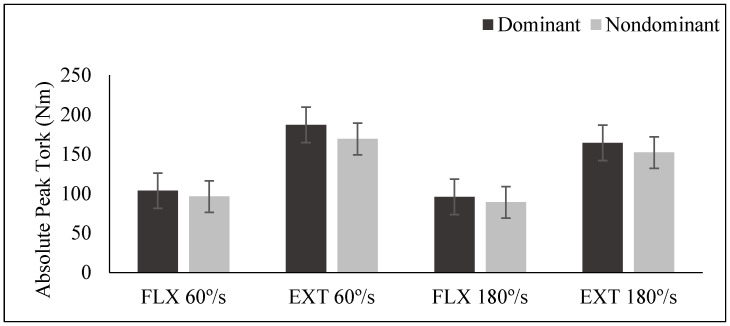
Inter-limb asymmetry and extension dominance in peak torque values of ski mountaineering sprint competitors.

**Figure 3 medicina-61-01237-f003:**
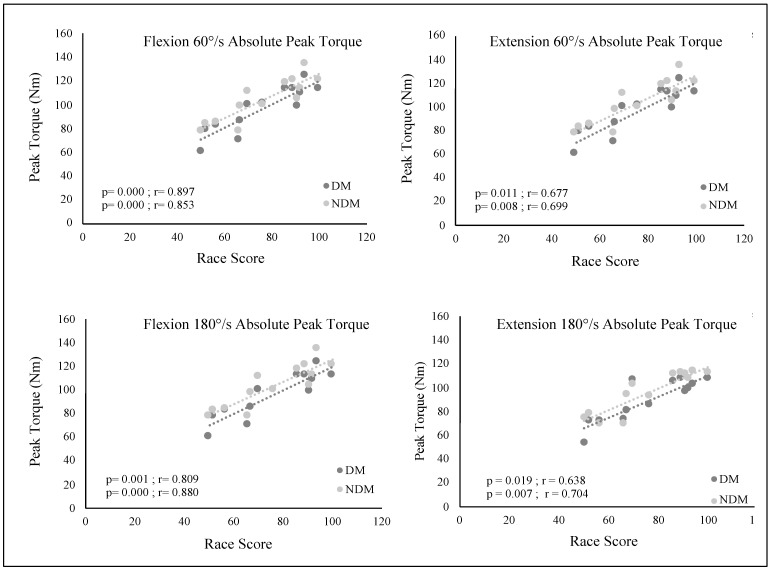
Correlation between flexor and extensor absolute peak torque values (Nm) in the dominant and non-dominant legs at 60°/s and 180°/s angular velocities, and competition performance scores.

**Table 1 medicina-61-01237-t001:** Characteristics and performance metrics of ski mountaineering sprint athletes.

Variables	Mean ± SD	Range
Age (years)	19.6 ± 2.12	16–22
Body weight (kg)	68.5 ± 2.69	65–73
Height (cm)	175.3 ± 5.19	165–182
BMI (kg/m^2^)	22.3 ± 1.27	21–25
Body fat (%)	8.6 ± 1.81	6.43–11.74
Race score	76.17 ± 17.04	50.20–100
Race time (s)	323.77 ± 78.60	235–468

BMI: body mass index.

**Table 2 medicina-61-01237-t002:** Dominant leg peak torque (60–180°/s) regression on performance.

Predictors	B	Std. Err.	β ^b^	95% CI		t	*p*-Value ^a^
(Constant)	−153.372	30.508		−228.023	−78.722	−5.027	0.002
Flexion at 60°/s	−78.510	32.542	−0.973	−158.136	1.117	−2.413	0.050
Extension at 60°/s	61.317	13.647	1.043	27.926	94.709	4.493	0.004
Flexion at 180°/s	226.710	41.531	2.566	125.088	328.333	5.459	0.002
Extension at 180°/s	−57.148	12.212	−1.332	−87.029	−27.268	−4.680	0.003

F = 3.978; Adjusted R^2^ = 0.498; *p* = 0.046; Durbin–Watson = 1.743; Maximum VIF = 1.826. Adj. R^2^: adjusted coefficient of determination. ^a^ *p*-values were calculated using the regression model. ^b^ β: unstandardized regression coefficients, CI: confidence interval.

**Table 3 medicina-61-01237-t003:** Non-dominant leg peak torque (60–180°/s) regression on performance.

Predictors	B	Std. Err.	Β ^b^	95%CI		t	*p*-Value ^a^
(Constant)	−40.651	14.671		−75.343	−5.959	−2.771	0.028
Flexion at 60°/s	−0.729	0.486	−0.729	−1.879	0.421	−1.499	0.177
Extension at 60°/s	0.251	0.140	0.526	−0.080	0.582	1.796	0.116
Flexion at 180°/s	1.755	0.506	1.468	0.559	2.951	3.469	0.010
Extension at 180°/s	−0.090	0.123	−0.194	−0.381	0.200	−0.737	0.485

F = 9.696; Adjusted R^2^ = 0.492; *p* = 0.002; Durbin–Watson = 2.145; Maximum VIF = 1.845. Adj. R^2^: adjusted coefficient of determination. ^a^ *p*-values were calculated using the regression model. ^b^ β: unstandardized regression coefficients, CI: confidence interval.

## Data Availability

All relevant data are within the manuscript. The dataset will be published on Dryad after acceptance of the manuscript, and DOI will be provided.

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
