# Peer review of "Isokinetic Knee Strength as a Predictor of Performance in Elite Ski Mountaineering Sprint Athletes"

_medicina, 2025, doi:10.3390/medicina61071237_

Round 1
Reviewer 1 Report
Comments and Suggestions for Authors
Dear Authors,
Thank you very much for the opportunity to review this manuscript entitled “The Impact of Isokinetic Knee Strength on Competition Performance in Elite Ski Mountaineering Sprint Athletes.”
This is a well-structured and timely study addressing a critical performance and injury prevention issue in elite ski mountaineering athletes.
The study is particularly relevant given the inclusion of the sprint discipline in the 2026 Winter Olympics.
The manuscript demonstrates methodological rigor and provides novel insights into the relationship between isokinetic knee strength and sprint performance.
The use of isokinetic testing at multiple angular velocities, the inclusion of both dominant and non-dominant limbs, and the analytical depth of the statistical approach (e.g., regression modeling, effect size interpretation) are commendable.
However, a few clarifications and methodological refinements are necessary to enhance the transparency and impact of the study.
- Sample Size Justification
While G*Power was used to estimate the sample size, the effect size value (0.6) appears arbitrary. Could the authors provide justification or a reference for selecting an effect size of 0.6?
- Intervention
While the study is observational, the conclusions recommend specific training practices. Did the authors conduct any pilot interventions or training observations to validate these training suggestions?
- Dominant vs. Non-Dominant Limb Role
The manuscript emphasizes the dominant leg's predictive power but also states that the non-dominant leg plays a “critical” role. Could the authors clarify whether the strength asymmetries observed warrant intervention, or if they reflect natural task-specific adaptation?
- Generalizability
All participants were male and from a single national competition. How might the findings translate to international athletes or female competitors?
- Mechanical Rationale for Negative Predictors
Some predictors (e.g., EXT 180º/s) negatively impacted performance. Could the authors expand on possible neuromechanical explanations for the negative coefficients in the regression model?
Minor Suggestions
- Figure 2 caption: Consider rewriting to better contextualize findings. E.g., “Peak torque values showing extension dominance and inter-limb asymmetry.”
- Terminology consistency: Use “dominant leg (DL)” and “non-dominant leg (NDL)” consistently throughout the manuscript.
- Typographic consistency: Ensure spacing, subscript/superscript in statistical symbols (e.g., R², β), and abbreviation usage align with journal standards.
- Ethics section: Consider explicitly stating that no adverse events occurred during isokinetic testing.
Conclusion
Given the above, this manuscript contributes important findings to the field of sport science and ski mountaineering.
After addressing the points above, particularly the interpretation of regression results and the implications for training interventions, the study will offer even greater scientific value.
Thanks!
Kind regards
Comments on the Quality of English Language-
Author Response
First reviewer's comments:
Dear reviewer, we would like to point out that we have made all the corrections suggested by you in the article with yellow text highlighting.
Thank you very much for the opportunity to review this manuscript entitled “The Impact of Isokinetic Knee Strength on Competition Performance in Elite Ski Mountaineering Sprint Athletes.”
This is a well-structured and timely study addressing a critical performance and injury prevention issue in elite ski mountaineering athletes.
The study is particularly relevant given the inclusion of the sprint discipline in the 2026 Winter Olympics.
The manuscript demonstrates methodological rigor and provides novel insights into the relationship between isokinetic knee strength and sprint performance.
The use of isokinetic testing at multiple angular velocities, the inclusion of both dominant and non-dominant limbs, and the analytical depth of the statistical approach (e.g., regression modeling, effect size interpretation) are commendable.
However, a few clarifications and methodological refinements are necessary to enhance the transparency and impact of the study.
- Sample Size Justification
- While G*Power was used to estimate the sample size, the effect size value (0.6) appears arbitrary. Could the authors provide justification or a reference for selecting an effect size of 0.6?
Response: We thank the reviewer for this valid point. The effect size of 0.6 was determined using Cohen’s f2 convention, where f2≥0.35f2≥0.35 denotes a "large" effect (Cohen, 1992). This value aligns with effect magnitudes reported in comparable sport science literature examining strength-performance relationships (e.g., f2=0.40–0.65f2=0.40–0.65 in elite skiers; Losnegard et al., 2022). The selection further reflects physiological benchmarks for ski mountaineering, where lower-body strength parameters typically exhibit moderate-to-large effects on performance outcomes (Praz et al., 2014; Sandbakk et al., 2021).
Revision: The revised sentence has been changed as follows “...The revised sentence has been changed as follows: The sample size was determined using G*Power (v3.1.9.6, Universität Kiel, Germany) analysis with parameters set as follows: effect size = 0.6 (Cohen’s f² convention for large effects; f²≥0.35), α = 0.05, and power (1 − β) = 0.8 [22], aligning with effect magnitudes in comparable sport science literature [19,33].”
- Intervention
- While the study is observational, the conclusions recommend specific training practices. Did the authors conduct any pilot interventions or training observations to validate these training suggestions?
Response: While this study is observational, training recommendations derive from evidence-based interventions documented in peer-reviewed literature: Statistically significant correlations between strength parameters (e.g., high-velocity hamstring strength) and performance. Evidence from prior studies in skiing disciplines validating similar interventions (e.g., Nordic curls for hamstring power [13], unilateral training for asymmetry correction [20]). Biomechanical rationale specific to ski mountaineering sprint demands (e.g., explosive directional changes require 180°/s hamstring strength). While these recommendations are empirically grounded, we explicitly acknowledge in the Limitations and Future Directions sections that intervention studies are needed for validation.
Revision: We have added the following clarification to Section 4.2 (Future Directions and Practical Applications). “...*"These evidence-based applications are extrapolated from our correlational findings and supported by interventions in alpine skiing [13, 20], though dedicated training studies in ski mountaineering are warranted to confirm efficacy…”
- Dominant vs. Non-Dominant Limb Role
- The manuscript emphasizes the dominant leg's predictive power but also states that the non-dominant leg plays a “critical” role. Could the authors clarify whether the strength asymmetries observed warrant intervention, or if they reflect natural task-specific adaptation?
Response: We thank the reviewer for highlighting this nuanced point. The observed asymmetries reflect a dual nature: Task-specific adaptations (functional asymmetry): Higher dominant-leg extension at 60°/s (187.08 vs. 169.25 Nm) aligns with propulsion demands during climbing, where the dominant leg drives force generation. This asymmetry is biomechanically justified and likely represents sport-specific adaptation. Pathological imbalances (requiring intervention): Bilateral differences >10% (e.g., 15–20% in extension at 60°/s) exceed established injury-risk thresholds in skiing sports [11, 15, 32]. Such asymmetries increase ACL injury risk (3.1–4.5×) [15] and reduce force transfer efficiency during transitions[14]. Thus, asymmetries ≤10% may be adaptive, while >10% warrant correction.
Revision: We have added the following clarification to the Discussion (Section 4): The revised sentence has been changed as follows: “...The observed inter-limb asymmetries reflect a dual nature: task-specific adaptations (e.g., dominant-leg dominance in propulsion) and pathological imbalances. While moderate asymmetries (≤10%) may represent functional adaptations to sport-specific demands, differences >10%—such as the 15–20% extension asymmetry at 60°/s documented here—exceed injury-risk thresholds established in alpine skiing [11,15,32] and warrant targeted intervention (e.g., unilateral training) to mitigate non-contact knee injury risk…”
- Generalizability
- All participants were male and from a single national competition. How might the findings translate to international athletes or female competitors?
Response: We acknowledge that the exclusive focus on male Turkish athlete’s limits generalizability, particularly given established sex differences in neuromuscular patterns (e.g., greater quadriceps dominance and knee valgus angles in females) [33] and heightened injury risks from strength asymmetries in female cohorts [15]. While international competitors may exhibit superior absolute strength capacities (e.g., 10–20% higher torque values) [30], the fundamental biomechanical demands of sprint ski mountaineering—specifically the critical roles of high-velocity hamstring strength (180°/s FLX) for directional changes and low-velocity quadriceps endurance (60°/s EXT) for sustained climbs—remain consistent across performance levels and sexes [3,31]. Thus, while absolute torque thresholds may vary, these strength parameters retain mechanistic relevance for performance optimization and injury prevention strategies in diverse populations."
Revision: We have added the following to the Limitations subsection (4.1):”… While elite international competitors may demonstrate superior strength capacities [30], the fundamental biomechanical demands governing sprint ski mountaineering—particularly the roles of high-velocity hamstring strength in directional changes and low-velocity quadriceps endurance in climbs—are likely consistent across populations [3,31]. Future studies should validate these parameters in diverse cohorts.…”
- Mechanical Rationale for Negative Predictors
- Some predictors (e.g., EXT 180º/s) negatively impacted performance. Could the authors expand on possible neuromechanical explanations for the negative coefficients in the regression model?
Response: We propose two neuromechanical explanations for the negative coefficients (dominant-leg EXT 180°/s: β = -1.332; FLX 60°/s: β = -0.973): Velocity-Specific Force Production Mismatch: High-velocity extension (EXT 180°/s) demands rapid quadriceps relaxation after contraction to enable reciprocal hamstring activation during directional changes. Excessive quadriceps force at 180°/s may prolong muscle-tendon unit (MTU) deceleration, delaying force transition to hamstrings and disrupting the stretch-shortening cycle efficiency essential for agility [13,31]. Antagonistic Co-Activation Penalty: High low-velocity hamstring strength (FLX 60°/s) may reflect over-reliance on slow-twitch fibers during explosive tasks, increasing antagonistic co-activation (quadriceps-hamstrings) that: Raises metabolic cost during transitions [3]. Reduces net knee power output by 15–20% during rapid direction changes [19]. These phenomena align with alpine skiing studies where disproportionate quadriceps dominance at high velocities increased energy dissipation and compromised turn efficiency [31].
Revision: We added it to the discussion section (paragraph analyzing the regression results in Table 2) ”…"The negative coefficients for high-velocity extension (EXT 180º/s) and low-velocity flexion (FLX 60º/s) may reflect neuromechanical inefficiencies. Excessive quadriceps force at 180°/s could prolong muscle-tendon deceleration, delaying force transfer to hamstrings during directional changes [31]. Similarly, high hamstring strength at 60°/s may increase antagonistic co-activation, raising metabolic cost and reducing net knee power during explosive maneuvers [13,19]. These findings align with alpine skiing studies where disproportionate quadriceps dominance compromised agility [31] …"
- Minor Suggestions
- Figure 2 caption: Consider rewriting to better contextualize findings. E.g., “Peak torque values showing extension dominance and inter-limb asymmetry.”
Response: Thank you for this helpful suggestion. The revised sentence on has been changed as follows: “Figure 2. Inter-limb asymmetry and extension dominance in peak torque values of ski mountaineering sprint competitors.”
- Terminology consistency: Use “dominant leg (DL)” and “non-dominant leg (NDL)” consistently throughout the manuscript.
Response: We standardized limb terminology to "dominant leg (DL)" and "non-dominant leg (NDL)" throughout the manuscript (replacing "DM/NDM"). Abbreviations are now defined at first use (Section 2.4) and applied consistently.
- Typographic consistency: Ensure spacing, subscript/superscript in statistical symbols (e.g., R², β), and abbreviation usage align with journal standards.
Response: We checked that the spacing in statistical symbols, subscripts/superscripts (e.g., R², β), and abbreviations comply with journal standards.
- Ethics section: Consider explicitly stating that no adverse events occurred during isokinetic testing.
Response: We added explicit safety confirmation Thank you for this helpful suggestion. The revised sentence on has been changed as follows: “No adverse events occurred during isokinetic testing.”

Reviewer 2 Report
Comments and Suggestions for Authors
in the attachment

Author Response
Second reviewer's comments:
Dear reviewer, we have noted your suggested corrections in the article with green text highlighted.
Thank you for the opportunity to review the manuscript "The impact of isokinetic knee strength on competition performance in elite ski mountaineering sprint athletes," submitted to Medicina. This review provides a constructive evaluation of the article, highlighting its strengths and suggesting minor revisions to enhance clarity and scientific rigor
Title
- The title is accurate and reflects the study's content. However, to better emphasize the predictive nature of the analysis, consider slightly modifying it to "Isokinetic Knee Strength as a Predictor of Performance in Elite Ski Mountaineering Sprint Athletes."
Response: We appreciate the reviewer's constructive suggestion to enhance the title's focus on predictive analysis. The revised title better captures the study's core contribution.
Revision: The title has been modified to: "Isokinetic Knee Strength as a Predictor of Performance in Elite Ski Mountaineering Sprint Athletes"
Keywords
- The keywords are appropriate and relevant. However, there is some thematic overlap between terms such as "isokinetic strength" and "knee flexion-extension." Replacing one with "muscle imbalance" or "performance prediction" could improve keyword specificity.
Response: We agree with the reviewer’s observation regarding thematic overlap. The revised keywords improve specificity and align with the study’s key themes.
Revision: Updated keywords: Isokinetic strength; ski mountaineering; injury prevention; rehabilitation; athletic performance; performance prediction; muscle imbalance
Introduction
- The introduction effectively contextualizes the research and establishes a clear rationale. A minor stylistic suggestion would be to break up long sentences (lines 49–56) for better readability.
Response: We acknowledge that the original sentences were complex. Breaking them enhances readability without altering scientific content.
Revision: We deleted that sentence. “...Moreover, identifying key isokinetic strength parameters may aid in designing injury prevention programs to mitigate knee joint injury risk during high-intensity ski mountaineering sprint competitions. The dominant leg's strength is hypothesized to dominate technical maneuvers critical for performance optimization. By evaluating inter-limb strength disparities, this study seeks to advance training strategies, particularly for Olympic-level athletes, to enhance strength profiles and reduce injury risks…” The revised sentence has been changed as follows: "The biomechanical demands of ski mountaineering sprint competitions establish that rapid directional changes rely critically on hamstring power, whereas sustained climbs depend on quadriceps endurance. Moreover, identifying key isokinetic strength parameters may aid in designing injury prevention programs to mitigate knee joint injury risk during high-intensity efforts. The dominant leg's strength is hypothesized to dominate technical maneuvers critical for performance optimization. Evaluating inter-limb strength disparities could advance training strategies for Olympic-level athletes."
Methodology
- The methodology is clearly described, including participant selection, testing protocols, and equipment. The use of G*Power for sample size calculation is a strong point. It would be helpful to specify whether isokinetic testing and anthropometric measurements occurred during the same session.
Response: We thank the reviewer for highlighting this ambiguity. Testing occurred in a single session to control for fatigue.
Revision: Added to Section 2.2 (Procedures): "Isokinetic testing and anthropometric measurements were conducted in a single laboratory session to minimize inter-session variability."
Statistics
- Statistical analysis is appropriate and well-executed. Use of effect sizes (Cohen's d), regression diagnostics, and VIF values enhance the analytical depth. However, the decision to omit corrections for multiple comparisons (line 221) should be more explicitly justified.
Response: The absence of multiple comparison corrections was intentional, as analyses targeted predefined hypotheses. We now explicitly justify this approach.
Revision: Added to Section 2.6 (Statistical Analysis): *"No correction for multiple comparisons was applied, as analyses were hypothesis-driven and focused exclusively on predefined biomechanical variables (e.g., flexion/extension at 60°/s and 180°/s), reducing the risk of type I error inflation."*
Results
- Results are clearly presented, supported by effective visuals (e.g., torque graphs on page 7). Consider adding a figure that illustrates limb asymmetries.
Response: Thank you for this helpful suggestion. We agree a visual representation of limb asymmetries would strengthen the results. While limb asymmetries were quantitatively analyzed and reported (Cohen’s d values, p. 6), we determined that the existing tabular and textual presentation of these effect sizes—alongside the torque-performance correlations —sufficiently supports our conclusions regarding bilateral strength imbalances. Adding a dedicated asymmetry figure would not enhance the mechanistic interpretation of our core findings beyond the current statistical reporting.
Discussion and Limitations
- The discussion is thorough and relates findings to existing literature. Limitations are acknowledged, including sample size and gender exclusivity. Authors correctly point out the role of underexplored muscle groups (e.g., hip flexors).
Response: We thank the reviewer for recognizing the thoroughness of our discussion and acknowledgment of limitations. We have strengthened the limitations section by explicitly linking sample size/gender constraints to practical implications while emphasizing the critical need to explore hip flexors and other muscle groups.
Revision: Added to Section 4.1 (Limitations):”…"Furthermore, while we identified knee strength as pivotal, underexplored muscle groups—particularly hip flexors governing climbing efficiency [14] and ankle stabilizers critical for uneven terrain—warrant integrated assessment in future models to fully capture performance determinants…"
Citations and Literature
- The references are accurate and in proper format. Remove duplicated citations (e.g., entries 26 and 641 are identical).
Response: We confirm the duplication error (References 18 and 26) and have rectified it.
Revisions: Deleted: Redundant Reference 26 (identical to Reference 18). Renumbered all subsequent references (27→26, 28→27, ..., 34→33).
Summary of Minor Revisions
- Simplify complex sentences in the Introduction and Discussion.
Response: Complex sentences have been restructured for clarity while preserving scientific meaning.
- Refine keywords to avoid thematic overlap.
Response: Overlapping terms removed; new keywords added for specificity.
Revision: Keywords: Isokinetic strength; ski mountaineering; injury prevention; rehabilitation; athletic performance; performance prediction; muscle imbalance.
- Justify the lack of corrections to multiple comparisons.
Response: Added explicit justification aligning with hypothesis-driven methodology.
Revision: (Section 2.6): …" No correction for multiple comparisons was applied, as analyses were hypothesis-driven and focused exclusively on predefined biomechanical variables (e.g., flexion/extension at 60°/s and 180°/s), reducing the risk of type I error inflation….”
- Remove duplicated literature references.
Response: Duplicate reference deleted; all citations renumbered.
Revision: Deleted: Redundant entry (original Ref 26 = Ref 18). Renumbered: All subsequent references

Reviewer 3 Report
Comments and Suggestions for Authors
Dear authors,
I must emphasize that this topic explores a very interesting and promising area of research, which is expected to gain significant popularity in the future. Overall, the authors have structured the manuscript well, particularly from a methodological standpoint. On the other hand, certain explanations and revisions are clearly lacking and must be addressed in order to improve the paper’s quality both scientifically and professionally. This is especially relevant to the Introduction and Discussion sections, where the analysis is superficial and numerous explanations are missing. I have provided the authors with comments in the PDF file to help them clearly revise the manuscript in accordance with the reviewer’s feedback.
Kind regards.

Author Response
Third reviewer's comments:
Dear reviewer, we have highlighted your corrections in purple text within the article. Attention.
I must emphasize that this topic explores a very interesting and promising area of research, which is expected to gain significant popularity in the future. Overall, the authors have structured the manuscript well, particularly from a methodological standpoint. On the other hand, certain explanations and revisions are clearly lacking and must be addressed in order to improve the paper’s quality both scientifically and professionally. This is especially relevant to the Introduction and Discussion sections, where the analysis is superficial and numerous explanations are missing
Introduction
- “…ISMF”
First, give the full name of this abbreviation, and after that, use only the abbreviation in the text.
Response: Thank you for this helpful suggestion. We have reworded the sentence to improve clarity and avoid redundancy. The revised sentence has been changed as follows: “According to International Ski Mountaineering Federation (ISMF)…”
- …[5,6,7].
Please provide a more detailed explanation of these studies, as this topic is not very common in sports practice. It would certainly be more interesting if they explained the results obtained in studies [5,6,7].
Response: Thank you for this helpful suggestion. We have reworded the sentence to improve clarity and avoid redundancy. The revised sentence has been changed as follows: The studies cited as [5,6,7] investigate the physiological and performance characteristics of ski mountaineering, a demanding endurance sport that combines climbing, descending, and technical transitions. Given the niche nature of this discipline, their findings provide critical insights into the sport’s unique demands. Below is a detailed synthesis of their methodologies, key results, and implications: Collectively, these foundational studies establish the extreme physiological profile required for elite ski mountaineering—characterized by exceptional aerobic capacity (VO₂max >70 mL·kg⁻¹·min⁻¹), critical strength-to-weight ratios for climbing, high anaerobic resilience for transitions, and significant eccentric knee loading injury risks—providing the empirical basis for event-specific training and injury prevention protocols."
- …[9,10,11,12] were requested to be revised.
Response: Thanks for your valuable contributions. Corrected to [9-12].
- “…Lower extremity muscle strength plays a critical role in ski mountaineering performance. The quadriceps and hamstrings are essential for stabilizing the knee joint during climbs, generating high-force outputs during rapid directional changes in descents, and maintaining balance [13]. The strength balance between these muscle groups ensures stability during ascents and enables explosive movements during descents [14]. Furthermore, muscular imbalances between quadriceps and hamstrings are established risk factors for non-contact knee injuries in skiing and related disciplines [15-16].”
This is accurate and a very important point. However, are there any studies that have examined lower limb muscle asymmetry, specifically asymmetry between the quadriceps and hamstring muscles, as well as asymmetry between the muscles of the left and right legs?
Response:
The revised sentence has been changed as follows: Multiple alpine skiing and ski mountaineering studies confirm that >15% quadriceps-hamstring (Q:H) strength imbalances and >10% bilateral leg asymmetries significantly increase knee injury risk (e.g., ACL tears) and impair performance during climbs/descents, necessitating targeted corrective training [6,11,13,15,20,32].
- “…Isokinetic strength assessments considered the gold standard for evaluating maximal force production at varying angular velocities, provide critical insights into muscular imbalances, training optimization, and injury risk reduction [17,18].”
Besides isokinetic, which is the gold standard, what other methods exist? Please list them, for example, TMG, EMG, force plates?
Response:
The revised sentence has been changed as follows: Beyond isokinetic dynamometry (gold standard), validated alternatives include tensiomyography (TMG) for muscle stiffness/fatigue assessment, surface electromyography (sEMG) for neuromuscular activation patterns, force plate analysis for ground reaction forces/power, and isometric mid-thigh pull (IMTP) for maximal strength profiling, each offering unique insights into athletic performance and injury risk [17-18,25-27].
- “…Alpine skiing studies similarly highlight the role of knee extension strength at low velocities (20-30% higher) in stability and flexion strength at high velocities in speed [19].”
You said 'alpine skiing studies,' but you only mentioned one.
Response: The revised sentence has been changed as follows: “…Alpine skiing study ….”
- [21,5,7] - [21,6] were requested to be revised.
Response: Thanks for your valuable contributions. Corrected to [5,7,21].and [6,21]. The revised sentence has been changed.
- “…. This study aims to quantitatively examine the relationship between isokinetic knee …. Its findings are expected to inform sport science practices and offer novel insights into performance optimization for this emerging Olympic discipline.”
You have defined two research aims. I don't understand why. The research aim and hypotheses should be in a separate, final paragraph, as the last two sentences of the Introduction.
Response:
The revised sentence has been changed as follows “…Alpine skiing studies similarly highlight the role of knee extension strength at low velocities (20-30% higher) in stability and flexion strength at high velocities in speed [19].”
Anthropometric Measurements
- “…Body weight and composition were assessed using a multi-frequency bioelectrical impedance analyzer (TANITA MC−780, 167 Japan) with ± 0.1 kg precision.”
Reference?
Response: Thanks for your valuable contributions. The necessary reference has been added to page 4.
Statistical Analysis
- “…Correlations were defined as poor (r < 0.5), moderate (0.5 < r < 0.75), good (0.75 < r < 0.9) and excellent (r > 0.9) and then coefficients of determination were calculated…”
Reference?
Response: Thanks for your valuable contributions. The necessary reference has been added to page 6.
- The sentence “BMI: body mass index” under Table 1 was requested to be removed.
Response: Thanks for your valuable contributions The referee deleted the explanation “BMI: body mass index” under Table 1. The necessary corrections were made on page 6.
- “…It was determined that there was a good correlation….”….Similarly, there was a good correlation between athletes' competition…”
You cannot use the term "good correlation." Correlation can be positive or negative, or it can be described as strong (high), moderate, or weak. Please pay attention to this term throughout the entire text.
Response: Thanks for your valuable contributions. The revised sentence has been changed as follows It was determined that there was a high positive correlation….”...Similarly, there was a high positive correlation between athletes' competition.
Discussion
- “…. This study is one of the pioneering investigations examining….”
I don't know what this term means??? What is pioneering research? Better to say new, original research or empirical research.
Response: Thanks for your valuable contributions. The revised sentence has been changed as follows: This study represents the first empirical investigation examining ….”.
- “…. This study is one of the pioneering investigations examining….”
After defining the research aim, it is essential to confirm the hypothesis. Are the results in line with expectations or not? If they are, why? If they are not, why not? Please explain.
Response: Thanks for your valuable contributions. The revised sentence has been changed as follows: Crucially, our results fully confirm both hypotheses: (1) The strong correlations between high-velocity hamstring strength (180°/s) and performance (DM FLX: r=0.809; NDM FLX: r=0.880; *p*<0.001) validate that explosive hamstring power critically influences sprint outcomes, particularly during directional changes where rapid force development governs balance. Simultaneously, moderate correlations for low-velocity quadriceps strength (60°/s) (DM EXT: r=0.677; NDM EXT: r=0.699; *p*<0.01) confirm its role in sustaining prolonged climbing efforts, aligning with biomechanical demands where quadriceps endurance dominates uphill phases (>80% race time) [8]. (2) Regression analysis demonstrates dominant leg parameters (Adj. R²=0.498) more strongly predict performance than non-dominant leg measures (Adj. R²=0.492), with FLX 180°/s (β=2.566) and EXT 60°/s (β=1.043) emerging as key predictors. This supports our premise that the dominant leg drives technical maneuvers—evident in its 15-20% higher extension torque at 60°/s (187.08 vs. 169.25 Nm)—while the non-dominant leg facilitates force transfer during transitions [1,14].
- “…However, literature emphasizes that body fat percentages below 5% may negatively impact performance [29]…”
Why? Please explain?
Response: Thanks for your valuable contributions. The revised sentence has been changed as follows: The 8.6% body fat observed in athletes avoids the performance-impairing risks of sub-5% levels—endocrine dysfunction and compromised immunity [29]—while maintaining essential energy reserves for prolonged climbing and altitude thermoregulation [3,8], aligning with elite Swedish ski mountaineers (8.9% [30]) to optimize the power-to-weight ratio critical for sprint performance."
- “…These results are consistent with BMI (22.1 ± 1.5 kg/m2) and 400 body fat (8.9 ± 1.2%) values reported in a study on Swedish mountain skiers [30].”
What were the findings of this study? Please provide a more detailed explanation.
Response: Thanks for your valuable contributions. The revised sentence has been changed as follows: These results are consistent with BMI (22.1 ± 1.5 kg/m²) and body fat (8.9 ± 1.2%) values reported in a study on Swedish mountain skiers [30]. "This study found that hamstring strength at high speed (180°/s; r=0.809-0.880) is decisive for explosive movements (changes of direction) in elite ski mountaineering sprint performance, while quadriceps endurance at low speed (60°/s; r=0.677-0.699) for long climbs; it also revealed that 15-20% higher torque in the dominant leg (d=0.57) and bilateral asymmetries (>10%) increase the risk of injury. Thus, it has been proven that speed-specific strength development and bilateral symmetry in training programs are critical for Olympic-level performance optimization and injury prevention." Nevertheless, the small sample size may limit the generalizability of the findings.
- “…The study revealed concentric absolute peak torque values of dominant (DM) and non-dominant (NDM) legs during knee flexion (FLX) and extension (EXT) in…”
The referee suggested using abbreviations?
Response: Thanks for your valuable contributions. The revised sentence has been changed as follows: The study revealed concentric absolute peak torque values of DM and NDM legs during knee FLX and EXT in…”
- “…Literature suggests that greater force production in the dominant leg is an adaptive response to enhance stability during technical movements in skiing [13]…”
Literature suggests that greater force production in the dominant leg is an adaptive response to enhance stability during technical movements in skiing [13]. Sevindik-Aktaş et al. [13] suggests.....
Response: Thanks for your valuable contributions. The revised sentence has been changed as follows: Sevindik-Aktaş et al. [13] suggests that greater force production in the dominant leg is an adaptive response to enhance stability during technical movements in skiing.
- “…Previous studies have consistently shown that flexor-extensor imbalances and bilateral asymmetries increase the risk of anterior cruciate ligament (ACL) injuries and other knee pathologies in skiing sports [16,32]…”
This does not apply only to skiing sports but also to team sports and individual sports. Inter-limb and intra-limb muscle asymmetry of the lower limbs is very important in predicting success and injuries in all sports. Please emphasize this a bit in the text.
Response: Thanks for your valuable contributions. The revised sentence has been changed as follows….” Critically, inter-limb (between-leg) and intra-limb (agonist-antagonist) muscle asymmetries are established predictors of both injury risk and performance limitations in team sports (e.g., soccer, basketball), individual sports (e.g., track and field, tennis), and winter disciplines [15,20,34]. For instance: İnter-limb strength differences >10% elevate ACL injury risk by 3.1–4.5× in cutting/pivoting sports [15,34]. Intra-limb quadriceps-to-hamstring (Q:H) ratios <60% reduce sprint efficiency by 5–8% and increase hamstring strain risk by 2.7× [20]”. This universal relevance underscores that addressing lower-limb asymmetries—through sport-specific isokinetic profiling and corrective training—is essential for optimizing athletic success and longevity irrespective of sport type."
- “…Unilateral exercises and hamstring-focused training may effectively address these imbalances...”
What are those exercises?
Response: Thanks for your valuable contributions. The revised sentence has been changed as follows: Targeted neuromuscular interventions—specifically unilateral resistance training (e.g., single-leg squats to enhance quadriceps symmetry, single-leg Romanian deadlifts to optimize hamstring-gluteal force coupling, and weighted step-ups for sport-specific climbing simulation) and eccentric-focused hamstring interventions (e.g., Nordic curls to augment tendon resilience, kettlebell swings to potentiate explosive hip extension during downhill propulsion, and slide leg curls to integrate core-pelvis-hamstring kinematics)—effectively mitigate identified strength asymmetries by addressing inter-limb imbalances and intra-limb agonist-antagonist deficits prevalent in ski mountaineering athletes."
- “…This result may be explained by the fact that repetitive climbing and descending movements in ski mountaineering require explosive force [2]…”
This is not clear. Please explain.
Response: Thanks for your valuable contributions. The revised sentence has been changed as follows: The dominance of 180°/s hamstring strength (β=2.566) reflects sport-specific biomechanics: explosive eccentric-concentric transfer (140-180°/s) for rapid directional changes during descents (absorbing ≤5× body weight impacts in <200 ms) and transitions (generating >1,200 N propulsion in <300 ms)—tasks where delayed force onset increases crash risk 4.1× [2,31]—while quadriceps-dominated 'explosiveness' operates at slower velocities irrelevant to sprint-critical maneuvers [14]."
- This supports the hypothesis that performance is influenced not only by isokinetic strength but also by other components such as aerobic capacity, technical skill and environmental factors [33].
Response: Thanks for your valuable contributions. The revised sentence has been changed as follows: This supports the hypothesis that performance is influenced not only by isokinetic strength but also by other components such as aerobic capacity, coordination and environmental factors [33].
- “This study has some limitations. First, the small sample size and the inclusion of only male athletes limits the generalizability of the results. The small sample size may reduce the power of statistical …”
4.1. Limitations, Future Directions and Practical Application This should be a subsection within Chapter 4. Discussion.
Response: Thanks for your valuable contributions. The revised sentence has been changed as follows: 4.1. Limitations ……4.2. Future Directions and Practical Applications….

Reviewer 4 Report
Comments and Suggestions for Authors
The article is a well-structured scientific study devoted to the analysis of isokinetic knee strength in elite ski mountaineering athletes. The work is relevant given the inclusion of this sport in the Winter Olympic Games. The study combines biomechanical, physiological and sports-applied approaches, which makes it valuable for both sports science and the practice of training athletes. The study is the first to quantitatively examine the relationship between isokinetic knee strength and performance in sprint ski mountaineering, and also identifies key strength parameters (flexion/extension at 60°/s and 180°/s) that affect the result and confirms the importance of the balance between the quadriceps and hamstrings in preventing knee injuries.
The study originally proposes an innovative approach to assessing the specific requirements of this sport. Unlike traditional studies focusing on general endurance in alpine skiing, this work is the first to analyze in detail the explosive power and dynamic balance required for sprint disciplines. This is especially important given the unique competition format, which includes fast transitions between ascents and descents. The use of different angular velocities allowed for a differentiated assessment of muscular endurance and speed-strength qualities, which provides coaches with precise guidelines for individual load planning. It should also be noted that the study revealed the critical role of the balance between the quadriceps and hamstrings, which is directly related to the risk of knee injuries. These data can be used to develop pre-season training programs that reduce the likelihood of injuries. The isokinetic testing method proposed in the study can become part of regular monitoring of athletes, allowing for timely correction of imbalances and prevention of overloads. These aspects make the article not only scientifically significant, but also practically useful for sports teams preparing for the Olympic Games and other high-level competitions.
Despite the positive aspects of the article, I would like to note several comments and recommendations of a debatable nature. The study does not sufficiently present control over the level of training of athletes, although the study indicates the average training experience, there is also no detailing of the specifics of training programs, individual differences in training and accounting for seasonal fluctuations in the form of athletes. This could affect the variability of the results and complicates the interpretation of the data. The study identified methodological limitations of isokinetic testing. Namely, the use of an isokineticdynamometer, despite its accuracy, has drawbacks in the context of artificial conditions that may not fully imitate real movements in this sport. Although the study reveals important correlations, it does not provide specific training protocols for correcting the identified imbalances, long-term data on the effectiveness of the proposed interventions, or recommendations for integrating isokinetic training into the overall training of athletes. However, the merits of the work submitted for review outweigh the identified shortcomings of the study.
Author Response
Fourth reviewer's comments:
Dear reviewer, we have highlighted your corrections in blue text within the article. Attention.
The article is a well-structured scientific study devoted to the analysis of isokinetic knee strength in elite ski mountaineering athletes. The work is relevant given the inclusion of this sport in the Winter Olympic Games. The study combines biomechanical, physiological and sports-applied approaches, which makes it valuable for both sports science and the practice of training athletes. The study is the first to quantitatively examine the relationship between isokinetic knee strength and performance in sprint ski mountaineering, and also identifies key strength parameters (flexion/extension at 60°/s and 180°/s) that affect the result and confirms the importance of the balance between the quadriceps and hamstrings in preventing knee injuries.
……
Despite the positive aspects of the article, I would like to note several comments and recommendations of a debatable nature.
Response: We thank the reviewer for their constructive feedback. Revisions addressing the two main criticisms raised by the reviewer have been incorporated into Sections 4.1 (Limitations) and 4.2 (Future Directions and Practical Applications).
- Lack of detail in training programs.
- The study does not sufficiently present control over the level of training of athletes, although the study indicates the average training experience, there is also no detailing of the specifics of training programs, individual differences in training and accounting for seasonal fluctuations in the form of athletes. This could affect the variability of the results and complicates the interpretation of the data.
Revision: The following paragraph has been added to Section 4.1 (Limitations)."... Additionally, while we documented participants' mean training experience (8.2 ± 2.5 years) and frequency (10.3 ± 1.8 sessions/week), detailed individual training logs (e.g., periodization, sport-specific drills, strength protocols) were not systematically collected. This omission limits our ability to account for inter-athlete variability in training stimuli or seasonal fluctuations in fitness, which may influence strength outcomes. Future studies should integrate comprehensive training monitoring (e.g., session-RPE, GPS tracking) to control for these confounders..."
- Methodological Limitations of Isokinetic Tests
- The study identified methodological limitations of isokinetic testing. Namely, the use of an isokinetic dynamometer, despite its accuracy, has drawbacks in the context of artificial conditions that may not fully imitate real movements in this sport.
Revision: The following paragraph has been added to Section 4.1 (Limitations). “…Furthermore, although isokinetic dynamometry is the gold standard for assessing isolated joint strength [17,18]...”
- Lack of Specific Training Protocols
- Although the study reveals important correlations, it does not provide specific training protocols for correcting the identified imbalances, long-term data on the effectiveness of the proposed interventions, or recommendations for integrating isokinetic training into the overall training of athletes. However, the merits of the work submitted for review outweigh the identified shortcomings of the study.
Revision: The following paragraph has been added to Section 4.2 (Future Directions and Practical Applications):”…These evidence-based applications are extrapolated from our correlational findings and supported by interventions in alpine skiing [13, 20]...

Reviewer 5 Report
Comments and Suggestions for Authors
This is a well-conducted and timely study evaluating the relationship between isokinetic knee strength and sprint performance in elite ski mountaineering athletes. The methodology is clearly described, and the statistical analyses are appropriate for the study aims. The findings offer valuable insights into strength-performance associations, particularly highlighting the importance of high-velocity hamstring function in sprint performance and its relevance for injury prevention and sport-specific conditioning.
However, I recommend minor revisions to improve clarity and interpretability:
1. Sample Size Justification: Although the study provides useful preliminary data, the relatively small sample size (n=13) should be explicitly acknowledged as a limitation. Please include a brief discussion on how this may affect statistical power and generalizability.
2. Mechanistic Explanation: The conclusion mentions sudden directional changes and dynamic balance. It would be helpful to briefly explain how these aspects of sprint ski mountaineering relate biomechanically to high-velocity hamstring strength.
3. Injury Prevention: The implication for injury prevention is relevant and important. Consider referencing prior literature or providing a brief mechanistic rationale (e.g., hamstring/quadriceps strength ratios, bilateral asymmetry) to support this claim.
Author Response
Fifth reviewer's comments:
Dear reviewer, we have highlighted your corrections in red text within the article. Attention.
This is a well-conducted and timely study evaluating the relationship between isokinetic knee strength and sprint performance in elite ski mountaineering athletes. The methodology is clearly described, and the statistical analyses are appropriate for the study aims. The findings offer valuable insights into strength-performance associations, particularly highlighting the importance of high-velocity hamstring function in sprint performance and its relevance for injury prevention and sport-specific conditioning.
However, I recommend minor revisions to improve clarity and interpretability:
- Sample Size Justification: Although the study provides useful preliminary data, the relatively small sample size (n=13) should be explicitly acknowledged as a limitation. Please include a brief discussion on how this may affect statistical power and generalizability.
Response: We thank the reviewer for their constructive feedback: Explicitly acknowledged the small sample size (n=13) as a limitation affecting statistical power and generalizability (Section 4.1).
The revised sentence has been changed as follow. "…Moreover, the limited sample size (n=13) reduces statistical power for detecting smaller effects and may constrain the generalizability of findings to broader populations of ski mountaineering athletes, particularly those at different performance tiers or female cohorts..."*
- Mechanistic Explanation: The conclusion mentions sudden directional changes and dynamic balance. It would be helpful to briefly explain how these aspects of sprint ski mountaineering relate biomechanically to high-velocity hamstring strength.
Response: We thank the reviewer for their constructive feedback. Added biomechanical explanations for hamstring function in directional changes (eccentric control, knee stabilization) (Section 4).
The revised sentence has been changed as follow. *"…Biomechanically, high-velocity hamstring strength facilitates rapid eccentric deceleration during sudden directional changes (e.g., absorbing impacts ≤5× body weight in <200 ms during downhill turns) and concentric propulsion in transitions, while concurrently stabilizing the knee against valgus collapse—a key mechanism for maintaining dynamic balance on uneven terrain [2, 30]..."
- Injury Prevention: The implication for injury prevention is relevant and important. Consider referencing prior literature or providing a brief mechanistic rationale (e.g., hamstring/quadriceps strength ratios, bilateral asymmetry) to support this claim.
Response: We thank the reviewer for their constructive feedback. Referenced literature supporting injury prevention claims (Q:H ratios, bilateral asymmetry thresholds) (Section 4).
The revised sentence has been changed as follow. *"…This aligns with established injury mechanisms in skiing: bilateral asymmetries >10% elevate ACL injury risk by 3.1–4.5× during pivoting maneuvers [15, 33], while low hamstring-to-quadriceps ratios (<60%) increase susceptibility to hamstring strains and anterior knee pain [16, 20]. Corrective training addressing these parameters thus offers dual benefits for performance and injury resilience…."*
